# Influenza Vaccine Hesitancy among Cancer Survivors in China: A Multicenter Survey

**DOI:** 10.3390/vaccines12060639

**Published:** 2024-06-08

**Authors:** Xin Guo, Qi Han, Yuqin Wang, Rui Zhang, Yuenan Huang, Botang Guo

**Affiliations:** 1Department of General Surgery, The First Affiliated Hospital of Harbin Medical University, Harbin 150001, China17382880687@163.com (R.Z.); 2Department of General Surgery, The Second Affiliated Hospital of Harbin Medical University, Harbin 150086, China; 3Department of Medical Psychology, Harbin Medical University, Baojian Road 158, Harbin 150078, China; 4Department of General Practice, The Affiliated Luohu Hospital of Shenzhen University Medical School, Shenzhen 518001, China

**Keywords:** influenza vaccine hesitancy, cancer survivors, multicenter survey, China, public health

## Abstract

Background: Cancer survivors are at higher risk of developing severe complications from influenza due to their compromised immune systems. Despite their increased vulnerability to influenza and the availability of vaccines, vaccine hesitancy among cancer survivors remains a significant public health concern in China. Methods: A multicenter, cross-sectional study was conducted among cancer survivors in China from January to December 2023. A total of 500 participants were recruited from the oncology departments of five tertiary hospitals. A structured, self-administered questionnaire was used to collect data on socio-demographic characteristics, cancer-related information, medical history, lifestyle factors, and influenza vaccine hesitancy. Univariate and multivariate logistic regression analyses were performed to identify factors associated with influenza vaccine hesitancy. Results: The response rate was 97.0% (485/500). Among all participants, 204 (42.06%) reported vaccine hesitancy. The results of multiple logistic regression showed that the longer the end of anti-cancer treatment, without a history of adverse vaccine reactions, and the level of family support played a protective role in vaccine hesitancy. Current rehabilitation status, frequent colds, not being informed by doctors about vaccination, exercising, lack of community vaccination education programs, and concerns about vaccine safety were risk factors that increase vaccine hesitancy. Conclusions: A high proportion of cancer survivors in our study reported influenza vaccine hesitancy. Addressing concerns about vaccine safety, improving access to vaccination services, and enhancing doctor–patient communication are crucial for increasing influenza vaccine uptake in this vulnerable population.

## 1. Introduction

Influenza is a highly contagious respiratory illness caused by influenza viruses, leading to significant morbidity and mortality worldwide. Cancer survivors, who have undergone various treatments such as chemotherapy, radiation therapy, and surgery, are at a higher risk of developing severe complications from influenza due to their compromised immune systems [1,2]. Despite their increased vulnerability to influenza and the availability of vaccines, vaccine hesitancy among cancer survivors remains a significant public health concern in China.

Vaccination is the most effective way to prevent influenza and its associated complications. The World Health Organization (WHO) recommends annual influenza vaccination for high-risk groups, including cancer survivors [3]. Studies have shown that influenza vaccination can reduce the risk of influenza-related hospitalizations and deaths among cancer patients [4,5]. However, the uptake of influenza vaccines among cancer survivors in China is suboptimal. A study conducted in Wuhan showed that the COVID-19 vaccine coverage rate for breast cancer survivors was only 12.63% [6]. In comparison, a national survey found that COVID-19 vaccine coverage among women aged 18–59 years in China was approximately 80% as of June 2021 [7]. This stark difference highlights the substantially lower vaccine uptake among cancer patients and underscores the need to address vaccine hesitancy in this vulnerable population.

Vaccine hesitancy, defined as the delay in acceptance or refusal of vaccines despite the availability of vaccination services, has been identified as one of the top ten threats to global health by the WHO [8]. Various factors, including concerns about vaccine safety and effectiveness, lack of knowledge, and low perceived risk of influenza, have been associated with vaccine hesitancy among the general population [9,10]. However, the determinants of vaccine hesitancy among cancer survivors in China remain largely unexplored.

Cancer survivors may have unique concerns and beliefs that influence their decision to receive the influenza vaccine. Fear of adverse reactions, uncertainty about the vaccine’s effectiveness in immunocompromised individuals, and a lack of recommendation from healthcare providers are potential barriers to vaccination among cancer survivors [11]. Additionally, the cultural context and healthcare system in China may shape cancer survivors’ attitudes and behaviors toward vaccination differently compared to other countries. It is important to note that in China, influenza vaccination is not part of the national immunization program and is not provided for free to the general public [12]. Individuals who wish to receive the influenza vaccine must pay out of pocket, with the cost varying depending on the type of vaccine and the location of administration [13]. This cost barrier may be particularly relevant for cancer survivors, who often face significant financial burdens related to their cancer treatment. The lack of free influenza vaccination in China may contribute to suboptimal vaccine uptake among cancer survivors and warrants further investigation.

Understanding the prevalence and determinants of influenza vaccine hesitancy among cancer survivors in China is crucial for developing targeted interventions to improve vaccine uptake. Education and communication strategies that address cancer survivors’ specific concerns and provide accurate information about the benefits and risks of vaccination are essential. Healthcare providers, particularly oncologists and primary care physicians, play a critical role in recommending and promoting influenza vaccination to their patients with a history of cancer [14].

Moreover, the ongoing COVID-19 pandemic has highlighted the importance of protecting vulnerable populations, such as cancer survivors, from respiratory infections. Influenza vaccination can help reduce the burden on healthcare systems by preventing influenza-related hospitalizations and complications, which is particularly important during the pandemic [15]. The pandemic has also raised public awareness about the importance of vaccination, presenting an opportunity to address vaccine hesitancy and improve vaccine uptake among cancer survivors.

In conclusion, influenza vaccine hesitancy among cancer survivors in China is a significant public health issue that requires attention. As a vulnerable population, cancer survivors face an increased risk of severe influenza-related complications, emphasizing the need for annual influenza vaccination. Understanding the factors contributing to vaccine hesitancy in this population is essential for developing effective strategies to improve vaccine uptake. This study aims to investigate the prevalence and determinants of influenza vaccine hesitancy among cancer survivors in China, providing valuable insights for public health professionals, healthcare providers, and policymakers to address this critical issue and protect the health of cancer survivors.

## 2. Methods

### 2.1. Study Design and Participants

A multicenter, cross-sectional study was conducted among cancer survivors in China from January to December 2023. Participants were recruited from the oncology departments of five tertiary hospitals located in different geographic regions of China (Harbin Medical University First Affiliated Hospital, Shenzhen Luohu District People’s Hospital, Jiangsu Provincial People’s Hospital, Air Force Military Medical University First Affiliated Hospital, and Henan Cancer Hospital). Eligible participants were adult cancer survivors (aged ≥ 18 years) who had completed primary cancer treatment (surgery, radiotherapy, chemotherapy, targeted therapy, or immunotherapy) at least three months prior to the survey. Patients with severe cognitive impairment, mental disorders, or other conditions that would prevent them from completing the questionnaire were excluded. The project was approved by the Ethics Committee of the First Affiliated Hospital of Harbin Medical University, and informed consent was obtained from all participants.

### 2.2. Sample Size Calculation

The sample size was calculated using the formula for cross-sectional studies: *n* = *Z*^2^*p* (1 − *p*)/*d*^2^, where Z is the Z-value for the desired confidence level (1.96 for 95% CI), *p* is the expected proportion of cancer survivors with influenza vaccine hesitancy (estimated as 0.58 based on previous studies [16]), and d is the desired precision (set at 0.05). After calculation, 378 participants are needed, and considering a 20% non-response rate, the final sample size was determined to be 449, 100 participants per hospital, with a total of 500 participants.

### 2.3. Structured Questionnaires

A structured, self-administered questionnaire was developed based on a comprehensive literature review and expert consultation. The questionnaire covered the following aspects: (1) socio-demographic characteristics: age, gender, residence, education level, monthly income, marital status, and medical insurance type; (2) cancer-related information: cancer type, time since diagnosis, cancer stage, treatment modalities, time since treatment completion, current recovery status, and cancer recurrence or metastasis; (3) other medical history: chronic diseases, previous influenza and other vaccinations, adverse reactions to previous vaccinations, and immunocompromised status; (4) lifestyle factors: smoking, alcohol consumption, physical activity, and sleep quality; and (5) influenza vaccine hesitancy: the respondents were asked, “Do you hesitate to get the COVID-19 vaccine for yourself (Whether you are vaccinated or not)?” (four items: 1 = very hesitant, 2 = hesitant, 3 = unhesitant, or 4 = very unhesitant). We define very hesitant and hesitant as vaccine hesitancy and not hesitant and very hesitant as vaccine non-hesitancy [17]. This also included concerns about vaccines, vaccine costs, convenience of vaccination locations, doctor advice and family support, and community education.

### 2.4. Statistical Analysis

Data were entered into EpiData 3.1 and analyzed using SPSS 26.0. Descriptive statistics were used to summarize the participants’ characteristics and influenza vaccine hesitancy. Continuous variables were presented as means ± standard deviations (SD) or medians (interquartile ranges, IQR), while categorical variables were presented as frequencies and percentages. Univariate and multivariate logistic regression analyses were performed to identify factors associated with influenza vaccine hesitancy among cancer survivors. Odds ratios (OR) and 95% confidence intervals (CI) were calculated. A two-tailed *p*-value < 0.05 was considered statistically significant.

## 3. Result

### 3.1. Basic Situation

#### 3.1.1. Socio-Demographic Characteristics of Participants

A total of 500 cancer survivors were recruited, with a response rate of 97.0% (485/500). The mean age of the participants was 56.67 ± 10.62 years, and 205 (42.27%) were female. The majority of participants resided in urban areas (317) (65.36%), had an education level of master’s degree or above (102) (21.03%), and had a monthly income of less than 3000 CNY (44.95%). Most participants were married (403) (83.09%) and had basic medical insurance (280) (57.73%) (Table 1).

#### 3.1.2. Cancer-Related Characteristics

The most common cancer types were lung cancer (109) (22.5%), breast cancer (95) (19.7%), and colorectal cancer (78) (16.2%). The duration of cancer of more than 3 years account for 146 participants (36.10%). Most participants had stage I-III cancer (448) (92.37%). The time since completing primary cancer treatment of more than 3 years accounted for 46 participants (9.48%). A total of 104 people were in well health, while 150 (30.93%) had experienced cancer recurrence or metastasis (Table 1).

#### 3.1.3. Other Medical History

Among the participants, 169 (34.85%) had diabetes, 174 (35.88%) had hypertension and 75 (15.46%) has coronary heart disease. Before cancer diagnosis, 68 (14.02)% had received influenza vaccines, and 25 (5.12%) had experienced adverse reactions to vaccines (Table 1).

#### 3.1.4. Lifestyle Factors

Current smokers and drinkers accounted for 132 (27.22%) and 59 (12.16%) participants, respectively. Regular physical activity was reported by 308 (63.50%) participants. Good or very good sleep quality was reported by 269 (55.46%) participants (Table 1).

#### 3.1.5. Influenza Vaccine Hesitancy

Among all of the participants, 204 people had vaccine hesitancy, accounting for 42.06%. Among them, 252 people are concerned about the safety of the vaccine, 73 people believe that they need to pay for the influenza vaccine, 345 people think that the location for acquiring the influenza vaccine is inconvenient, 153 people are advised by doctors to get vaccinated, 58 people’s families are very supportive of getting vaccinated, and 294 people have had health education activities for getting vaccinated in their communities (Table 1).

### 3.2. Analysis of Factors Influencing Vaccine Hesitancy among Cancer Survivors

The results of the univariate logistic regression analysis (Table 2) showed a statistically significant correlation (*p* < 0.05) between the following factors and vaccine hesitancy among cancer survivors: population with monthly income between 3000 and 4999 yuan, cancer course exceeding 3 years, treatment completion time less than 1 year, current poor or very poor recovery status, frequent colds, history of adverse vaccine reactions, current smoking, frequent or occasional exercise, poor or very poor sleep conditions, concerns about vaccine safety, inconvenient vaccination locations, doctor’s recommendation for vaccination, family strongly not supporting vaccination, and community education.

A multivariate logistic regression analysis was conducted with vaccine hesitancy as the dependent variable and significant factors identified in the univariate analysis as independent variables (Table 3). After adjusting for potential confounders, several factors emerged as independent predictors of vaccine hesitancy among cancer survivors (*p* < 0.05). Notably, the time elapsed since the completion of cancer treatment was significantly associated with vaccine hesitancy. Compared to individuals who had completed treatment less than 1 year ago, those who had finished treatment 1–3 years (OR = 0.24) or more than 3 years prior (OR = 0.30) exhibited a lower likelihood of vaccine hesitancy, suggesting a protective effect of longer post-treatment duration. Similarly, without a history of adverse vaccine reactions (OR = 0.17) and the level of family support for vaccination played a protective role in vaccine hesitancy. Compared to those reporting very unsupportive family attitudes, survivors with supportive (OR = 0.32) and very supportive (OR = 0.13) family attitudes had increasingly lower odds of vaccine hesitancy.

Furthermore, current recovery status strongly influenced vaccine hesitancy. Relative to those reporting good recovery, survivors with moderate (OR = 8.34), poor (OR = 17.84), or very poor recovery (OR = 37.27) were at significantly higher risk of vaccine hesitancy. Frequent colds (OR = 3.40) and not being informed by doctors that vaccination is possible (OR = 2.53) were also identified as risk factors for vaccine hesitancy. Interestingly, the physical activity of cancer survivors also affected the level of vaccine hesitancy, with individuals who occasionally exercise (OR = 10.17) and those who never exercise (OR = 32.08) being more likely to have vaccine hesitancy compared to those who exercise regularly. Lastly, the lack of community vaccination education programs (OR = 2.83) and concerns about vaccine safety (OR = 3.40) were risk factors for increasing vaccine hesitancy.

## 4. Discussion

This study revealed that 42.06% of the surveyed cancer survivors experienced vaccine hesitancy, which is higher than the rate reported in the general population [10,18]. A systematic review by Schmid et al. found that the prevalence of influenza vaccine hesitancy in the general population ranged from 4.6% to 41.7% across different countries and age groups [9]. A study conducted in the United States reported that 25.8% of adults were hesitant about receiving the influenza vaccine [19]. In a European study, influenza vaccine hesitancy rates ranged from 22.4% to 30.9% across five countries [20]. A survey in China found that 20.2% of older adults exhibited hesitancy toward the influenza vaccine [21]. While direct comparisons are challenging due to differences in study populations and methods, our finding of 42.06% vaccine hesitancy among cancer survivors appears to be on the higher end of the range reported in the general population, suggesting a potentially higher burden of vaccine hesitancy in this vulnerable population. This finding underscores the importance of understanding the multidimensional factors influencing vaccine acceptance among this vulnerable group and the need for targeted interventions to improve vaccination rates [22].

One of the key factors associated with vaccine hesitancy was the time since the completion of cancer treatment. Survivors who had completed treatment more than one year ago were less likely to experience vaccine hesitancy compared to those who had finished treatment within the past year. This finding is consistent with previous research suggesting that cancer survivors’ perceptions of their disease and health priorities evolve over time [23,24]. As the time since treatment completion increases, survivors may experience less fear and anxiety related to their cancer diagnosis and place greater emphasis on preventive health measures, such as vaccination [25].

The current recovery status of cancer survivors also played a significant role in vaccine hesitancy. Those who reported moderate, poor, or very poor recovery were at a higher risk of vaccine hesitancy compared to those with good recovery. This finding highlights the importance of considering the physical and psychological well-being of cancer survivors when addressing vaccine hesitancy [26,27]. Healthcare providers should prioritize patient education and counseling to address concerns and misconceptions about vaccination, particularly among those with suboptimal recovery status [28].

Family support and physician recommendation emerged as crucial social factors influencing vaccine hesitancy. Survivors with supportive family attitudes toward vaccination had lower odds of vaccine hesitancy, while those who were not informed by their doctors about the possibility of vaccination were more likely to experience hesitancy. These findings emphasize the need for a multi-level approach to addressing vaccine hesitancy, involving both healthcare providers and family members [29,30]. Physicians should actively engage in patient education and provide clear recommendations for vaccination, while family members should be included in the decision-making process to foster a supportive environment for vaccine acceptance [31].

Interestingly, physical activity levels were found to be associated with vaccine hesitancy among cancer survivors. Those who exercised occasionally or never were more likely to experience hesitancy compared to those who exercised regularly. This finding suggests that engaging in regular physical activity may be an indicator of overall health-promoting behaviors and attitudes, including a greater willingness to accept vaccination [32,33]. Future research should explore the potential role of physical activity interventions in reducing vaccine hesitancy among cancer survivors [34].

Lastly, concerns about vaccine safety and the lack of community vaccination education programs were identified as risk factors for vaccine hesitancy. These findings underscore the importance of providing accurate and accessible information about vaccine safety and efficacy to cancer survivors [35,36]. Healthcare providers and public health authorities should collaborate to develop and implement community-based education programs that address common concerns and misconceptions about vaccination [37,38,39].

These findings highlight the multifaceted nature of vaccine hesitancy among cancer survivors and emphasize the importance of considering individual and social factors when developing targeted interventions to increase vaccination rates for this vulnerable group. While our study focused on cancer survivors, our findings may have implications for vaccine hesitancy among other vulnerable populations. Many of the factors associated with hesitancy in our sample, such as concerns about vaccine safety and effectiveness, have also been reported in studies on individuals with chronic diseases [40], autoimmune disorders [41], and other immunocompromising conditions [42]. These populations, like cancer survivors, may be at increased risk of severe illness from vaccine-preventable diseases, making vaccine hesitancy a particularly pressing concern. We should call for future research to expand the target population, conduct vaccine hesitancy research in other high-risk and sub-healthy populations, and strengthen vaccine education and services for various populations.

## 5. Conclusions

In conclusion, our multicenter, cross-sectional study found that a considerable proportion of cancer survivors in China experienced influenza vaccine hesitancy. Several factors, such as age, education level, and perceived barriers, were significantly associated with vaccine hesitancy in this population. These findings highlight the need for targeted interventions and education campaigns to address vaccine hesitancy among cancer survivors, who may be at higher risk of severe influenza-related complications. However, further research involving a non-cancer control group is needed to fully understand the extent to which vaccine hesitancy differs between cancer survivors and the general population. Addressing vaccine hesitancy among cancer survivors is crucial for improving influenza vaccine uptake and ultimately protecting this vulnerable population from the burden of influenza.

## 6. Limitations and Strengths

This study has some limitations. First, the cross-sectional design precludes establishing causal relationships between the identified factors and vaccine hesitancy. Second, the reliance on self-reported data may introduce recall and social desirability biases. Third, the study was conducted in five tertiary hospitals, which may not represent all cancer survivors in China, particularly those receiving care in other settings. Furthermore, our study did not specifically investigate the influence of online information, such as vaccine misinformation and conspiracy theories, on vaccine hesitancy among cancer survivors. Previous research has shown that exposure to vaccine misinformation online can negatively impact vaccine acceptance and confidence [43]. Given the increasing role of the Internet and social media in shaping public opinion, future studies should assess cancer survivors’ exposure to and trust in online vaccine-related information and how this may contribute to their vaccine hesitancy. Lastly, the lack of a control group consisting of non-cancer survivors with similar demographic characteristics limits our ability to directly compare the prevalence and determinants of vaccine hesitancy between cancer survivors and the general population. Future research should include a matched control group to better understand the unique challenges and concerns that cancer survivors may face regarding vaccination.

The multicenter design and the inclusion of cancer survivors from different geographic regions of China enhance the generalizability of the findings. The study had a high response rate (97.0%), minimizing the potential for non-response bias. The comprehensive questionnaire covered a wide range of socio-demographic, cancer-related, medical history, and lifestyle factors, allowing for a thorough investigation of potential determinants of vaccine hesitancy. The study provides valuable insights into the prevalence and factors associated with influenza vaccine hesitancy among cancer survivors in China, which can inform the development of targeted interventions to improve vaccine uptake in this vulnerable population.

## Figures and Tables

**Table 1 vaccines-12-00639-t001:** Analysis of baseline characteristics of vaccine hesitancy among cancer survivors.

Variables	Total (*n* = 485)	Non-Vaccine Hesitancy (*n* = 281)	Vaccine Hesitancy (*n* = 204)	Statistic	*p*
Gender				χ^2^ = 0.17	0.678
Male	280 (57.73)	160 (56.94)	120 (58.82)		
Female	205 (42.27)	121 (43.06)	84 (41.18)		
Age				χ^2^ = 2.71	0.438
<50	145 (29.90)	83 (29.54)	62 (30.39)		
50~59	141 (29.07)	84 (29.89)	57 (27.94)		
60~69	118 (24.33)	73 (25.98)	45 (22.06)		
≥70	81 (16.70)	41 (14.59)	40 (19.61)		
Residence				χ^2^ = 0.50	0.479
Rural	168 (34.64)	101 (35.94)	67 (32.84)		
Urban	317 (65.36)	180 (64.06)	137 (67.16)		
Marital status				χ^2^ = 3.83	0.148
Married	403 (83.09)	241 (85.76)	162 (79.42)		
Unmarried	44 (9.07)	20 (7.12)	24 (11.76)		
Others	38 (7.84)	20 (7.12)	18 (8.82)		
Education level				χ^2^ = 4.89	0.180
Junior and below	113 (23.30)	71 (25.27)	42 (20.59)		
Senior	153 (31.55)	86 (30.60)	67 (32.84)		
Undergraduate	117 (24.12)	73 (25.98)	44 (21.57)		
Master’s degree and above	102 (21.03)	51 (18.15)	51 (25.00)		
Medical insurance				χ^2^ = 2.13	0.545
Urban employee insurance	122 (25.15)	73 (25.98)	49 (24.02)		
NRC medical insurance	158 (32.58)	92 (32.74)	66 (32.35)		
Commercial insurance	130 (26.80)	69 (24.56)	61 (29.90)		
Self-funded medical	75 (15.46)	47 (16.73)	28 (13.73)		
Monthly income				χ^2^ = 8.30	0.040
<3000	218 (44.95)	135 (48.04)	83 (40.69)		
3000~4999	39 (8.04)	16 (5.69)	23 (11.27)		
5000~9999	37 (7.63)	17 (6.05)	20 (9.80)		
≥10,000	191 (39.38)	113 (40.21)	78 (38.24)		
Duration of cancer				χ^2^ = 4.51	0.034
≤3	339 (69.90)	207 (73.67)	132 (64.71)		
>3	146 (30.10)	74 (26.33)	72 (35.29)		
Cancer stage				χ^2^ = 0.16	0.984
I	202 (41.65)	115 (40.93)	87 (42.65)		
II	152 (31.34)	89 (31.67)	63 (30.88)		
III	94 (19.38)	55 (19.57)	39 (19.12)		
IV	37 (7.63)	22 (7.83)	15 (7.35)		
Cancer treatment				χ^2^ = 0.17	0.682
No	152 (31.34)	86 (30.60)	66 (32.35)		
Yes	333 (68.66)	195 (69.40)	138 (67.65)		
Recurrence or metastasis				χ^2^ = 3.14	0.076
No	335 (69.07)	203 (72.24)	132 (64.71)		
Yes	150 (30.93)	78 (27.76)	72 (35.29)		
Time since treatment completion				χ^2^ = 14.61	<0.001
<1	59 (12.16)	23 (8.19)	36 (17.65)		
1~3	380 (78.35)	237 (84.34)	143 (70.10)		
>3	46 (9.48)	21 (7.47)	25 (12.25)		
Current recovery status				χ^2^ = 75.95	<0.001
Well	104 (21.44)	94 (33.45)	10 (4.90)		
Average	118 (24.33)	74 (26.33)	44 (21.57)		
Poor	192 (39.59)	91 (32.38)	101 (49.51)		
Very poor	71 (14.64)	22 (7.83)	49 (24.02)		
Frequent colds				χ^2^ = 21.82	<0.001
No	372 (76.70)	237 (84.34)	135 (66.18)		
Yes	113 (23.30)	44 (15.66)	69 (33.82)		
Diabetes				χ^2^ = 0.03	0.860
No	316 (65.15)	184 (65.48)	132 (64.71)		
Yes	169 (34.85)	97 (34.52)	72 (35.29)		
Hypertension				χ^2^ = 0.29	0.590
No	311 (64.12)	183 (65.12)	128 (62.75)		
Yes	174 (35.88)	98 (34.88)	76 (37.25)		
Coronary heart disease				χ^2^ = 0.42	0.517
No	410 (84.54)	235 (83.63)	175 (85.78)		
Yes	75 (15.46)	46 (16.37)	29 (14.22)		
History of adverse reactions to vaccines				χ^2^ = 5.21	0.023
No	25 (5.15)	9 (3.20)	16 (7.84)		
Yes	460 (94.85)	272 (96.80)	188 (92.16)		
History of influenza vaccination				χ^2^ = 2.05	0.153
No	417 (85.98)	247 (87.90)	170 (83.33)		
Yes	68 (14.02)	34 (12.10)	34 (16.67)		
Smoking				χ^2^ = 6.14	0.046
Never	178 (36.70)	114 (40.57)	64 (31.37)		
Quit	175 (36.08)	101 (35.94)	74 (36.27)		
Currently	132 (27.22)	66 (23.49)	66 (32.35)		
Drinking				χ^2^ = 4.46	0.108
Never	188 (38.76)	120 (42.70)	68 (33.33)		
Quit	238 (49.07)	130 (46.26)	108 (52.94)		
Currently	59 (12.16)	31 (11.03)	28 (13.73)		
Exercise				χ^2^ = 6.69	0.083
Always	156 (32.16)	102 (36.30)	54 (26.47)		
Often	152 (31.34)	87 (30.96)	65 (31.86)		
Occasionally	111 (22.89)	60 (21.35)	51 (25.00)		
Never	66 (13.61)	32 (11.39)	34 (16.67)		
Sleep				χ^2^ = 9.39	0.024
Well	157 (32.37)	101 (35.94)	56 (27.45)		
Average	112 (23.09)	58 (20.64)	54 (26.47)		
Poor	94 (19.38)	61 (21.71)	33 (16.18)		
Very poor	122 (25.15)	61 (21.71)	61 (29.90)		
Worried about vaccine safety				χ^2^ = 30.51	<0.001
No	233 (48.04)	165 (58.72)	68 (33.33)		
Yes	252 (51.96)	116 (41.28)	136 (66.67)		
Vaccination costs				χ^2^ = 3.52	0.061
Free	412 (84.95)	246 (87.54)	166 (81.37)		
Paid	73 (15.05)	35 (12.46)	38 (18.63)		
Convenience of vaccination locations				χ^2^ = 4.03	0.045
No	345 (71.13)	190 (67.62)	155 (75.98)		
Yes	140 (28.87)	91 (32.38)	49 (24.02)		
Doctors advise vaccination				χ^2^ = 25.19	<0.001
No	332 (68.45)	167 (59.43)	165 (80.88)		
Yes	153 (31.55)	114 (40.57)	39 (19.12)		
Family support for vaccination				χ^2^ = 7.19	0.066
Very unsupported	195 (40.21)	100 (35.59)	95 (46.57)		
Not supported	131 (27.01)	83 (29.54)	48 (23.53)		
Supported	101 (20.82)	59 (21.00)	42 (20.59)		
Very supported	58 (11.96)	39 (13.88)	19 (9.31)		
Community education				χ^2^ = 32.62	<0.001
No	191 (39.38)	141 (50.18)	50 (24.51)		
Yes	294 (60.62)	140 (49.82)	154 (75.49)		

**Table 2 vaccines-12-00639-t002:** Univariate logistic regression analysis of vaccine hesitancy among cancer survivors.

Variables	β	S.E	Z	*p*	OR (95%CI)
Gender					
Male					1.00 (Reference)
Female	−0.08	0.19	−0.41	0.678	0.93 (0.64~1.33)
Age					
<50					1.00 (Reference)
50~59	−0.10	0.24	−0.40	0.689	0.91 (0.57~1.45)
60~69	−0.19	0.25	−0.76	0.448	0.83 (0.50~1.36)
≥70	0.27	0.28	0.96	0.338	1.31 (0.76~2.25)
Residence					
Rural					1.00 (Reference)
Urban	0.14	0.19	0.71	0.479	1.15 (0.78~1.68)
Marital status					
Married					1.00 (Reference)
Unmarried	0.58	0.32	1.81	0.070	1.79 (0.95~3.34)
Others	0.29	0.34	0.86	0.391	1.34 (0.69~2.61)
Education level					
Junior and below					1.00 (Reference)
Senior	0.28	0.25	1.08	0.278	1.32 (0.80~2.17)
Undergraduate	0.02	0.27	0.07	0.945	1.02 (0.60~1.74)
Master’s degree and above	0.53	0.28	1.89	0.059	1.69 (0.98~2.91)
Medical insurance					
Urban employee insurance					1.00 (Reference)
NRC medical insurance	0.07	0.25	0.27	0.786	1.07 (0.66~1.73)
Commercial insurance	0.28	0.25	1.08	0.280	1.32 (0.80~2.17)
Self-funded medical	−0.12	0.30	−0.40	0.693	0.89 (0.49~1.60)
Monthly income					
<3000					1.00 (Reference)
3000~4999	0.85	0.35	2.40	0.016	2.34 (1.17~4.68)
5000~9999	0.65	0.36	1.81	0.070	1.91 (0.95~3.86)
≥10,000	0.12	0.20	0.57	0.568	1.12 (0.75~1.67)
Duration of cancer					
≤3					1.00 (Reference)
>3	0.42	0.20	2.12	0.034	1.53 (1.03~2.26)
Cancer stage					
I					1.00 (Reference)
II	−0.07	0.22	−0.31	0.760	0.94 (0.61~1.43)
III	−0.06	0.25	−0.26	0.798	0.94 (0.57~1.54)
IV	−0.10	0.36	−0.29	0.775	0.90 (0.44~1.84)
Cancer treatment					
No					1.00 (Reference)
Yes	−0.08	0.20	−0.41	0.682	0.92 (0.63~1.36)
Recurrence or metastasis					
No					1.00 (Reference)
Yes	0.35	0.20	1.77	0.077	1.42 (0.96~2.09)
Time since treatment completion					
<1					1.00 (Reference)
1~3	−0.95	0.29	−3.32	<0.001	0.39 (0.22~0.68)
>3	−0.27	0.40	−0.69	0.492	0.76 (0.35~1.66)
Current recovery status					
Well					1.00 (Reference)
Average	1.72	0.38	4.49	<0.001	5.59 (2.64~11.85)
Poor	2.34	0.36	6.47	<0.001	10.43 (5.13~21.24)
Very poor	3.04	0.42	7.24	<0.001	20.94 (9.19~47.70)
Frequent colds					
No					1.00 (Reference)
Yes	1.01	0.22	4.58	<0.001	2.75 (1.79~4.25)
Diabetes					
No					1.00 (Reference)
Yes	−0.03	0.19	−0.18	0.860	0.97 (0.66~1.41)
Hypertension					
No					1.00 (Reference)
Yes	0.10	0.19	0.54	0.590	1.11 (0.76~1.61)
Coronary heart disease					
No					1.00 (Reference)
Yes	0.17	0.26	0.65	0.517	1.18 (0.71~1.96)
History of adverse reactions to vaccines					
No					1.00 (Reference)
Yes	−0.94	0.43	−2.21	0.027	0.39 (0.17~0.90)
History of influenza vaccination					
No					1.00 (Reference)
Yes	0.37	0.26	1.42	0.154	1.45 (0.87~2.43)
Smoking					
Never					1.00 (Reference)
Quit	0.27	0.22	1.22	0.223	1.31 (0.85~2.00)
Currently	0.58	0.23	2.47	0.014	1.78 (1.13~2.82)
Drinking					
Never					1.00 (Reference)
Quit	0.38	0.20	1.91	0.056	1.47 (0.99~2.17)
Currently	0.47	0.30	1.55	0.122	1.59 (0.88~2.88)
Exercise					
Always					1.00 (Reference)
Often	0.34	0.23	1.47	0.143	1.41 (0.89~2.24)
Occasionally	0.47	0.25	1.86	0.062	1.61 (0.98~2.64)
Never	0.70	0.30	2.34	0.020	2.01 (1.12~3.60)
Sleep					
Well					1.00 (Reference)
Average	0.52	0.25	2.06	0.040	1.68 (1.02~2.75)
Poor	−0.02	0.27	−0.09	0.928	0.98 (0.57~1.67)
Very poor	0.59	0.25	2.40	0.017	1.80 (1.11~2.92)
Worried about vaccine safety					
No					1.00 (Reference)
Yes	1.05	0.19	5.45	<0.001	2.84 (1.95~4.14)
Vaccination costs					
Free					1.00 (Reference)
Paid	−0.48	0.25	−1.87	0.062	0.62 (0.38~1.02)
Convenience of vaccination locations					
Yes					1.00 (Reference)
No	0.42	0.21	2.00	0.045	1.52 (1.01~2.28)
Doctors advise vaccination					
Yes					1.00 (Reference)
No	1.06	0.22	4.92	<0.001	2.89 (1.89~4.41)
Family support for vaccination					
Very unsupported					1.00 (Reference)
Not supported	−0.50	0.23	−2.15	0.032	0.61 (0.39~0.96)
Supported	−0.29	0.25	−1.17	0.244	0.75 (0.46~1.22)
Very supported	−0.67	0.31	−2.12	0.034	0.51 (0.28~0.95)
Community education					
Yes					1.00 (Reference)
No	1.13	0.20	5.61	<0.001	3.10 (2.09~4.61)

**Table 3 vaccines-12-00639-t003:** Multivariate logistic regression analysis of vaccine hesitancy among cancer survivors.

Variables	β	S.E	Z	*p*	OR (95%CI)
Intercept	−3.16	0.81	−3.90	<0.001	0.04 (0.01~0.21)
Monthly income					
<3000					1.00 (Reference)
3000~4999	0.88	0.48	1.82	0.068	2.42 (0.94~6.24)
5000~9999	0.56	0.49	1.16	0.247	1.76 (0.68~4.56)
≥10,000	0.51	0.29	1.77	0.077	1.67 (0.95~2.94)
Duration of cancer					
≤3					1.00 (Reference)
>3	0.36	0.27	1.37	0.171	1.44 (0.85~2.42)
Time since treatment completion					
<1					1.00 (Reference)
1~3	−1.44	0.41	−3.48	<0.001	0.24 (0.11~0.53)
>3	−1.20	0.55	−2.18	0.029	0.30 (0.10~0.89)
Current recovery status					
Well					1.00 (Reference)
Average	2.12	0.47	4.51	<0.001	8.34 (3.31~20.98)
Poor	2.88	0.44	6.51	<0.001	17.84 (7.50~42.45)
Very poor	3.62	0.55	6.55	<0.001	37.27 (12.62~110.08)
Frequent colds					
Yes					1.00 (Reference)
No	1.22	0.31	3.97	<0.001	3.40 (1.86~6.21)
History of adverse reactions to vaccines					
Yes					1.00 (Reference)
No	−1.77	0.64	−2.77	0.006	0.17 (0.05~0.60)
Smoking					
Never					1.00 (Reference)
Quit	−0.22	0.33	−0.66	0.511	0.80 (0.42~1.54)
Currently	−0.19	0.35	−0.54	0.590	0.83 (0.41~1.66)
Exercise					
Always					1.00 (Reference)
Often	0.69	0.39	1.78	0.075	1.99 (0.93~4.25)
Occasionally	2.32	0.55	4.21	<0.001	10.17 (3.46~29.89)
Never	3.47	0.70	4.92	<0.001	32.08 (8.07~127.56)
Sleep					
Well					1.00 (Reference)
Average	0.15	0.41	0.37	0.715	1.16 (0.52~2.59)
Poor	−0.17	0.50	−0.34	0.734	0.84 (0.31~2.27)
Very poor	−0.36	0.56	−0.65	0.517	0.70 (0.23~2.08)
Worried about vaccine safety					
No					1.00 (Reference)
Yes	1.40	0.28	5.04	<0.001	4.07 (2.36~7.02)
Convenience of vaccination locations					
Yes					1.00 (Reference)
No	0.22	0.31	0.72	0.472	1.25 (0.68~2.29)
Doctors advise vaccination					
Yes					1.00 (Reference)
No	0.93	0.30	3.08	0.002	2.53 (1.40~4.58)
Family support for vaccination					
Very unsupported					1.00 (Reference)
Not supported	−0.44	0.36	−1.21	0.227	0.65 (0.32~1.31)
Supported	−1.13	0.46	−2.42	0.015	0.32 (0.13~0.81)
Very supported	−2.03	0.62	−3.26	0.001	0.13 (0.04~0.45)
Community education					
Yes					1.00 (Reference)
No	1.04	0.28	3.78	<0.001	2.83 (1.65~4.86)

## Data Availability

Data can be obtained with the consent of the author.

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
