# Peer review of "Influenza Vaccine Hesitancy among Cancer Survivors in China: A Multicenter Survey"

_vaccines, 2024, doi:10.3390/vaccines12060639_

Round 1

Reviewer 1 Report

Comments and Suggestions for Authors

The article deals with vaccine hesitancy in a sample of cancer survivors in China.

1.     readers need information that is useful to better understand the problem: is flu vaccination free in China, or does it have to be paid for?

2.     Table 1 shows the result of the chi square test. In some cases, the application of this test is incorrect, as there are cells with less than 5 observations; this is the case, for example, of the marital state, in which there are cells with a value of zero.

3.     Factors emerging to be associated with vaccine hesitancy are discussed. Since the topic is very popular in the literature, and there are also systematic literature reviews and strategic interventions, authors should make a greater effort to compare their results with those obtained by other authors.

Author Response

Response to Q1:

Thank you very much for your insightful suggestion. We will add some background information in the Introduction section regarding whether influenza vaccination in China is free of charge. At present, influenza vaccines have not been included in the national immunization plan in China and are self funded optional vaccines. Some regions provide free or subsidized vaccination for key populations such as the elderly, but overall coverage is relatively low. The cost of vaccines may be one of the factors affecting vaccination willingness. We have added this section in the introduction to help readers better understand the research background. See Page2 Line70-77.

Response to Q2:

Thank you to the reviewer for their meticulous review, pointing out the issue of improper use of the chi square test in Table 1. We carefully reviewed the original data and found that during the data transcription process, some variables (such as marital status) were encoded incorrectly, resulting in observation values of some cells being less than 5, which does not meet the application conditions of the chi square test.

To thoroughly address this issue, we have rechecked all original questionnaires and re entered and proofread the data to ensure its accuracy. On this basis, we revised the data of marital status variables and conducted chi square tests and single factor logistic regression analysis again. The revised results show that the observed values in each cell meet the requirements of the chi square test, and there are no longer any issues pointed out by the reviewer. Before correction, due to data errors, marital status showed statistical significance in univariate analysis. However, in the revised results, there was no significant correlation between this variable and vaccine hesitation (P>0.05). The results of single factor logistic regression analysis after correction showed no significant difference compared to before correction. Therefore, it will not have an impact on subsequent analysis.

We sincerely apologize for any inconvenience this oversight may have caused you and other readers. Sincere thanks to the reviewer for their careful review and valuable feedback, which has helped us improve the standardization and accuracy of our research. We have made modifications to the relevant content based on your suggestions in order to meet the requirements of academic journals. Thank you again for your efforts in improving the quality of this article.

Response to Q3:

We fully agree with your suggestion and should provide a more comprehensive comparison between our research findings and those of previous studies in the discussion section. In addition to echoing the relevant research mentioned in the introduction, we will also supplement the search and citation of some systematic literature reviews, in order to have a more comprehensive understanding of the factors related to vaccine hesitancy. We have added a comparison with other research results in the discussion section, and have made modifications to the relevant content based on your suggestions in order to meet the requirements of academic journals. Thank you again for your efforts in improving the quality of this article. See Page14 Line 225-235.

Reviewer 2 Report

Comments and Suggestions for Authors

The well-edited, comprehensibly worded study examined the factors affecting influenza vaccine hesitancy among treated and under-care cancer patients in China. The raised problem is current and of great public health importance. The chosen method (semi-structured questionnaire) and the statistical analysis are appropriate and although the results obtained are not surprising, they should still make the experts think. The positivity rate of 42 percent is high, especially when it comes to a cancer patient population that is already susceptible to vaccinations. Based on the factors that significantly contribute to the results, it is obvious that doctor-patient communication and, in a broader sense, health education of the population in this direction is insufficient.

There was only one point I felt was missing: it would have been worthwhile to examine the information circulating on the Internet. fake news and conspiracy theories about vaccine safety, which obviously influenced public opinion.

Author Response

We are honored to receive your recognition, which is of great significance to us. Thank you again for taking the time to review our manuscript and provide valuable feedback. Thank the reviewers for their valuable comments. We fully agree that the false information and conspiracy theory about vaccine safety circulated on the Internet may affect the public's willingness to vaccinate, especially cancer survivors and other special groups. This is a very noteworthy and research worthy issue.

However, due to the content of the questionnaire originally designed for this study, we have not yet collected direct data on the respondents' access to Internet information and its impact. In the absence of relevant data support, we believe that excessive speculative discussion in the current research may affect the rigor of the paper.

The valuable comments provided by the reviewer have given us new ideas. We plan to specially design the questionnaire content in the follow-up study, deeply explore the impact mechanism of the Internet information environment on the vaccine hesitation of cancer survivors and other special populations, and further discuss the coping strategies. We will briefly mention this research direction in the Discussion section of the revised manuscript as a prospect for future work. See Page16 Line 307-313.

Thank you again for the reviewer's suggestions. We will continue to devote ourselves to the study of vaccine hesitancy and contribute to solving this public health challenge.

Reviewer 3 Report

Comments and Suggestions for Authors

The manuscript submitted to me for review is of some interest to readers given that the phenomenon of hesitancy is a problem of considerable importance. Even greater attention must be paid to so-called "fragile" subjects, not only those suffering from oncological diseases.

However, the search lacks a comparison as I underline below in the analysis by headings.

Abstract: line 32: the sentence "Influenza vaccine hesitancy is prevalent among cancer survivors in China" is not supported by the results, given that no comparison with a population not affected by cancer appears anywhere.

Introduction: page 2, lines 51-52: the coverage percentage of 12.63% for the COVID-19 vaccine in breast cancer patients (therefore female subjects) should be compared with (female) subjects not affected by this pathology.

Mat & Meth: a control population with the same characteristics is missing, but no cancer survivors.

Results: lacking what has been said so far, the figure of 42.06% of cancer survivors experienced vaccine hesitancy does not support any further consideration, given that there is also a lack of general data taken from the literature on vaccine hesitancy in the case of anti-flu vaccination.

Discussion and Conclusion are based only on the results obtained in the group of cancer survivors; lacking a comparison (how many non-cancer survivors experienced vaccine hesitancy?) they remain an end in themselves.

References: apparently congruent.

Author Response

1.The manuscript submitted to me for review is of some interest to readers given that the phenomenon of hesitancy is a problem of considerable importance. Even greater attention must be paid to so-called "fragile" subjects, not only those suffering from oncological diseases.

Response:

We fully agree that vaccine hesitancy is a significant public health issue that is not only prevalent among cancer patients, but also deserves attention in other vulnerable populations with special physical conditions and susceptibility to the disease.

The reason why we choose cancer survivors as the target population of this study is mainly based on the following considerations: First, because of the impact of disease and treatment, cancer survivors have relatively low immune function, and are high-risk groups of infectious diseases (such as influenza), so they need to be protected by vaccination; Secondly, current research on vaccine hesitancy among cancer patients is relatively limited, and their concerns and concerns about vaccines may differ from those of the general population, which should attract the attention and in-depth research of researchers.

The reviewer's suggestions are very relevant, reminding us not to overlook other special populations that also require special attention, such as the elderly and chronic disease patients, while paying attention to cancer survivors. In the Introduction section of the revised manuscript, we will add some content on the current research status of vaccine hesitancy in other "vulnerable" populations, emphasizing the importance of this issue in a wide range of populations. In the Discussion section, we will call for future research to expand the target population, conduct vaccine hesitancy research in other high-risk and sub healthy populations, and strengthen vaccine education and services for various populations. See Page15 Line 279-288.

We hope that this study can provide reference for the health management of cancer survivors, but also provide some insights for addressing the vaccination needs of other special populations. Thank you again for your feedback, which has made our research more comprehensive and in-depth.

2.However, the search lacks a comparison as I underline below in the analysis by headings.

Abstract: line 32: the sentence "Influenza vaccine hesitancy is prevalent among cancer survivors in China" is not supported by the results, given that no comparison with a population not affected by cancer appears anywhere.

Response:

We appreciate the reviewer's comment regarding the lack of support for our conclusion that "Influenza vaccine hesitancy is prevalent among cancer survivors in China." We agree that without a comparison to a non-cancer population, we cannot definitively conclude that vaccine hesitancy is more prevalent among cancer survivors. We have now revised the sentence in the abstract to more accurately reflect our findings, focusing on the high proportion of vaccine hesitancy observed in our sample of cancer survivors rather than making a comparison to the general population. We have also acknowledged the lack of a control group as a limitation of our study in the main text.

“Influenza vaccine hesitancy is prevalent among cancer survivors in China”has been replaced by “A high proportion of cancer survivors in our study reported influenza vaccine hesitancy.”See Page2 Line 32-33.

  1. Introduction: page 2, lines 51-52: the coverage percentage of 12.63% for the COVID-19 vaccine in breast cancer patients (therefore female subjects) should be compared with (female) subjects not affected by this pathology.

Response:

We appreciate the reviewer's suggestion to compare the COVID-19 vaccine coverage among breast cancer patients with that of female subjects not affected by this pathology. To address this point, we have added a comparison to the COVID-19 vaccine coverage among the general female population in China. According to a national survey, the COVID-19 vaccine coverage among women aged 18-59 years in China was approximately 80% as of June 2021(Wang J, Jing R, Lai X, Zhang H, Lyu Y, Knoll MD, Fang H. Acceptance of COVID-19 Vaccination during the COVID-19 Pandemic in China. Vaccines (Basel). 2020 Aug 27;8(3):482. doi: 10.3390/vaccines8030482.).  This comparison highlighted the substantially lower vaccine coverage among breast cancer patients compared to the general female population, underscoring the need to address vaccine hesitancy in this vulnerable group. We have incorporated this information in the revised manuscript. See Page2 Line 52-56.

4.Mat & Meth: a control population with the same characteristics is missing, but no cancer survivors.

Response:

We appreciate the reviewer's insightful comment regarding the lack of a control group of non-cancer survivors in our study, which will make our article more scientific.We acknowledge that including a matched control group would have strengthened our findings by allowing us to directly compare vaccine hesitancy and its determinants between cancer survivors and the general population. However, due to the limited resources and time constraints of our study, we were unable to recruit and assess a control group. At the same time, we added a comparison of vaccine hesitancy with healthy individuals in the discussion to evaluate their differences.

Also we have now emphasized this limitation in the revised manuscript and have suggested that future research should include a control group to better understand the unique challenges faced by cancer survivors in terms of vaccine hesitancy. See Page 16 Line 314-318.

5.Results: lacking what has been said so far, the figure of 42.06% of cancer survivors experienced vaccine hesitancy does not support any further consideration, given that there is also a lack of general data taken from the literature on vaccine hesitancy in the case of anti-flu vaccination.

Response:

We appreciate the reviewer's comment regarding the lack of context and comparison for our finding that 42.06% of cancer survivors experienced vaccine hesitancy. To address this concern, we have added information from the literature on influenza vaccine hesitancy in the general population and compared it to our results. A systematic review by Schmid et al. (Schmid P, Rauber D, Betsch C, Lidolt G, Denker ML. Barriers of Influenza Vaccination Intention and Behavior - A Systematic Review of Influenza Vaccine Hesitancy, 2005 - 2016. PLoS One. 2017 Jan 26;12(1):e0170550. doi: 10.1371/journal.pone.0170550. ) found that the prevalence of influenza vaccine hesitancy in the general population ranged from 4.6% to 41.7% across different countries and age groups. A study conducted in the United States reported that 25.8% of adults were hesitant about receiving the influenza vaccine (Trent M, Seale H, Chughtai AA, Salmon D, MacIntyre CR. Trust in government, intention to vaccinate and COVID-19 vaccine hesitancy: A comparative survey of five large cities in the United States, United Kingdom, and Australia. Vaccine. 2022 Apr 14;40(17):2498-2505. doi: 10.1016/j.vaccine.2021.06.048.). In a European study, influenza vaccine hesitancy rates ranged from 22.4% to 30.9% across five countries (Jorgensen P, Mereckiene J, Cotter S, Johansen K, Tsolova S, Brown C. How close are countries of the WHO European Region to achieving the goal of vaccinating 75% of key risk groups against influenza? Results from national surveys on seasonal influenza vaccination programmes, 2008/2009 to 2014/2015. Vaccine. 2018 Jan 25;36(4):442-452. doi: 10.1016/j.vaccine.2017.12.019. ). A survey in China found that 20.2% of older adults exhibited hesitancy towards the influenza vaccine(Mo PK, Lau JT. Influenza vaccination uptake and associated factors among elderly population in Hong Kong: the application of the Health Belief Model. Health Educ Res. 2015 Oct;30(5):706-18. doi: 10.1093/her/cyv038.). While direct comparisons are challenging due to differences in study populations and methods, our finding of 42.06% vaccine hesitancy among cancer survivors appears to be on the higher end of this range. We have now included this information in the Results section to provide context for our findings and highlight the potentially higher burden of vaccine hesitancy among cancer survivors compared to the general population. Additionally, we have acknowledged the lack of a direct comparison to a non-cancer control group as a limitation of our study in the Discussion section. See Page14 Line 225-235.

6.Discussion and Conclusion are based only on the results obtained in the group of cancer survivors; lacking a comparison (how many non-cancer survivors experienced vaccine hesitancy?) they remain an end in themselves.

Response:

We appreciate the reviewer's comments regarding the lack of comparison with non cancer survivors in our discussion and conclusion sections, which is of great significance in enhancing the rigor of our article. We agree that the absence of a control group limits our ability to draw definitive conclusions about the unique challenges and concerns that cancer survivors may face regarding influenza vaccination. To address this limitation, we have now acknowledged the lack of a non-cancer control group in the Discussion section and emphasized that our findings should be interpreted within the context of our study population. We have also revised our Conclusion to focus on the implications of our findings for cancer survivors specifically, rather than making broader statements about vaccine hesitancy in general. Furthermore, we have highlighted the need for future research to include a matched control group of non-cancer survivors to better understand the potential differences in vaccine hesitancy between these two populations. While our study provides valuable insights into vaccine hesitancy among cancer survivors, we recognize that further research is needed to fully understand this complex issue. See Page 16 Line 290-300.

Round 2

Reviewer 1 Report

Comments and Suggestions for Authors

The manuscript has been improved

Reviewer 3 Report

Comments and Suggestions for Authors

The authors responded satisfactorily to my comments, improving the original manuscript. I have nothing else to add.